# Discovery of Screening Biomarkers for Major Depressive Disorder in Remission by Proteomic Approach

**DOI:** 10.3390/diagnostics11030539

**Published:** 2021-03-17

**Authors:** Hyebin Choi, Sora Mun, Eun-Jeong Joo, Kyu Young Lee, Hee-Gyoo Kang, Jiyeong Lee

**Affiliations:** 1Department of Senior Healthcare, Graduate School, Eulji University, Seongnam 13135, Korea; chb1996@naver.com (H.C.); sora6456@naver.com (S.M.); 2Department of Neuropsychiatry, School of Medicine, Eulji University, Daejeon 34824, Korea; jej1303@eulji.ac.kr (E.-J.J.); lky@eulji.ac.kr (K.Y.L.); 3Department of Psychiatry, Uijeongbu Eulji Medical Center, Eulji University, Gyeonggi 11759, Korea; 4Department of Psychiatry, Eulji General Hospital, Seoul 01830, Korea; 5Department of Biomedical Laboratory Science, College of Health Science, Eulji University, Seongnam 13135, Korea; 6Department of Biomedical Laboratory Science, College of Health Science, Eulji University, Uijeongbu 11759, Korea

**Keywords:** major depressive disorder, prothrombin, biomarker, proteomics, LC-MS/MS

## Abstract

Major depressive disorder (MDD) is a common disorder involving depressive mood and decreased motivation. Due to its high heterogeneity, novel biomarkers are required to diagnose MDD. In this study, a proteomic method was used to identify a new MDD biomarker. Using sequential window acquisition of all theoretical mass spectra acquisitions and multiple reaction monitoring analysis via mass spectrometry, relative and absolute quantification of proteins in the sera was performed. The results of the relative quantitation by sequential window acquisition for all theoretical mass spectra data showed that seven proteins were significantly differently expressed between MDD patients and other patients with remission status. However, absolute quantification by multiple reaction monitoring analysis identified prothrombin as the only significantly upregulated protein in the depressive state compared to remission (*p* < 0.05) and was, thus, subsequently selected as an MDD biomarker. The area under the curve for prothrombin was 0.66. Additionally, increased prothrombin/thrombin induced hyper-activation of platelets via activating protease-activated receptors, a feature associated with MDD; specifically, activated platelets secrete various molecules related to MDD, including brain-derived neurotropic factors and serotonin. Therefore, prothrombin is a potential screening, prognostic, and diagnostic marker for MDD.

## 1. Introduction

Major depressive disorder (MDD) is the most common mood disorder, and involves depressive moods and decreased motivation. In addition, it is a highly heterogeneous disease that causes various cognitive, mental, and physical symptoms, resulting in deterioration in daily functions. According to a study assessing the global burden of disease, injuries, and risk factors, depression, which previously held the 19th longest disability-adjusted life years for all diseases in 2016, reached the 13th position in 2019 [1,2]. One of the primary causes for the increased disease duration is the prescription of treatments prior to fully understanding their pathogenic mechanisms. Hence, ~40–50% of patients do not respond to their prescribed antidepressants. Although the etiology of depression is complex, one of the primary causes is believed to be a decrease in serotonin or defective serotonin receptors [3,4]. However, the mechanism underlying serotonin reduction in patients suffering from depression remains unknown. Considering that post-mortem studies, as well as studies of serotonin metabolites in body fluids (plasma and cerebrospinal fluid) have yielded inconsistent results [5,6,7,8,9,10,11], the pathogenesis of depression remains unclear, thus highlighting the importance of identifying biomarkers that can be readily monitored throughout treatment.

MDD is a highly heterogeneous disease with varying etiologies, presentations, processes, and responses to treatment [12]. It occurs idiopathically, with changes in the brain that are particularly difficult to observe compared to that in other organs. Hence, the complexity of the disease has prevented clear conclusions from being made regarding its etiology. Currently, the known causative risk factors include stressful events, cancer, and endocrine abnormalities [13]. Although several studies have identified specific genes associated with MDD, genetic factors only account for 30–40% of cases, while the remaining 60–70% of cases have non-genetic etiologies related to individual environmental factors, including gene-environment interactions [14]. Due to the various etiologies and disease complexities, including the varying susceptibility to episodes among patients, samples must be individually analyzed [15]. Additionally, considering that proteins are the final products of translation and can, therefore, reflect post-translational events, proteomic studies have the potential to detect non-genetic disease states [16,17,18,19].

When assessing protein content in samples, certain mass spectrometry strategies target specific proteins, whereas others do not [20]. Among the non-targeted strategies, shotgun proteomics represent a bottom-up approach in which a mass spectrometer is coupled with high-performance liquid chromatography (HPLC) to identify unknown proteins in a complex mixture [21]. Shotgun analysis is a powerful method in terms of high throughput and protein coverage [22]. Meanwhile, the sequential window acquisition of all theoretical mass spectra (SWATH) method was recently introduced to continuously fragment all peptides in the *m/z* window separated by a data independent acquisition-mass spectrometry (DIA-MS) mode. The resulting transition ions match the spectral library, which can be used for peptide and protein identification and quantification [23]. The SWATH method relies heavily on the spectral library and is analyzed by the shotgun proteomics method when creating the library [22]. SWATH is advantageous as it maximizes the peptides observed in individual samples, as well as in the entire sample set, to extend the proteome coverage range, reduce quantitative variability through experimental efficiency, and minimize missing data [24]. Therefore, SWATH is a strategic method for analyzing individual samples and was used in this study to identify screening markers for MDD.

Many researchers have sought to identify MDD biomarkers. The primary peripheral biomarkers that were identified are associated with inflammation and the immune response, such as C-reactive protein, cytokines, erythrocyte sedimentation rate, and tryptophan catabolites along the indoleamine-(2,3)-dioxygenase pathway. Additionally, biomarkers related to oxidative stress and antioxidant defense, such as malondialdehyde, 8-hydroxy-2-deoxyguanosine, and superoxide dismuatases, have also been reported [25]. However, variables such as the sex, age, and body mass index (BMI) of subjects can influence the reliability of results, and few studies have considered these variables in their analyses. For example, a meta-analysis evaluated biomarkers including gastrointestinal, immunological, and neurotrophic factors, and concluded that only cortisol was a potential marker of MDD [26]. However, the pool of participants included in this study was too heterogeneous to perform effective comparisons. Additionally, cortisol is influenced by the disease state. Hence, in the present study, we used sera from patients with depression as well as those in remission, all of whom were being administered antidepressants at the time of sample collection. Variables, such as gender, age, and BMI were also considered while identifying the MDD biomarkers. Comparison of sera protein abundance from patients with depression and those with remission status can reveal screening and prognostic biomarkers.

## 2. Materials and Methods

### 2.1. Patients and Sample Collection

Patients with MDD were recruited from the Eulji University Hospital, Daejeon, South Korea, and the approval to conduct the study was obtained from the institutional review board (EMC 2016-03-019, 31 March 2016). The number of patients in depressive and remission states were 22 and 20 in the discovery set and 47 and 40 in the validation set, respectively. Clinical information on the subjects is listed in Table 1. One subject from the discovery set with depressive status, and one patient from the validation set with remission status, did not have available BMI information. Two subjects from each group in the discovery and validation sets had no available smoking information. Figure 1 shows a schematic representation of the experiment design. Blood samples were collected from each subject in plain vacutainers. The serum was isolated by centrifugation at 4000× *g* for 5 min and stored at −80 °C until analysis. Each serum sample was collected to an equal volume of 1 µℓ to prepare pooled samples for preparing the peptide library.

### 2.2. Sample Preparation

#### 2.2.1. Depletion of Highly Abundant Proteins

To eliminate highly abundant proteins from the serum, including albumin, IgG, IgA, antitrypsin, haptoglobin, and transferrin, multiple-affinity removal system liquid chromatography (LC) column-human 6 (human 6-HC, 4.6 × 50 mm; Agilent Technologies, Santa Clara, CA, USA) was used. Samples were injected into the multiple-affinity removal system LC column to deplete high abundant proteins, while low-abundance proteins were eluted simultaneously. Using a Nanosep filter (Pall Nanosep, Ann Arbor, MI, USA) with a molecular weight cut-off of 3 kDa, the eluted solution was concentrated according to the manufacturer’s instructions.

#### 2.2.2. Peptide Digestion

The protein concentration of each serum sample was measured by performing Bradford assay (Bio-Rad, Hercules, CA, USA). For mass analysis, 100 µg of serum protein was prepared. For protein reduction, 5 mM Tris(2-carboxyethyl) phosphine (Pierce, Rockford, IL, USA) was treated at 37 °C, 300 rpm, for 30 min. Alkylation was performed by treatment with 15 mM iodoacetamide (Sigma-Aldrich, St. Louis, MO, USA) at 25 °C and 300 rpm for 1 h in the dark. The proteins were treated with mass spectrometry-grade trypsin gold (Promega, Madison, WI, USA) at 37 °C, 800 rpm, for 15 h (overnight) to generate peptides. The residual chemical reagent was eliminated using a C18 cartridge (Waters, Milford, MA, USA).

### 2.3. LC-MS/MS Analysis

#### 2.3.1. Information-Dependent Acquisition (IDA) and Data Analysis

A 12-well fractionator (3100 OFFGEL Low Red Kit, pH 3–10; Agilent Technologies) was used to separate proteins according to their isoelectric points, following the manufacturer’s protocol. This method was used to identify as many peptides as possible while preparing the library. A TripleTOF 5600 mass spectrometer (AB Sciex, Concord, Ontario, Canada) coupled with an Eksigent NanoLC 400 system and cHiPLC^®®^ spectrometer (AB Sciex) was used for IDA analysis. In each run, 2 µL sample solution (0.5 µg/µL) was loaded onto an Ekigent ChromXP NanoLC trap column (350 µm i.d. × 0.5 mm, ChromXP C18 3 µm) at a flow rate of 5000 nL/min for 5 min. The injected sample was eluted from the Eksigent Chrom XP NanoLC column (75 µm i.d. × 15 cm), at a flow rate of 300 nL/min for 120 min with a gradient of 5–90% mobile phase B. The following LC gradient was used: (time/mobile phases B%) 0 min/5%, 10.5 min/40%, 111.5 min/90%, 112 min/5%, and 120 min/5%. Mobile phase B was 90% acetonitrile/0.1% formic acid in HPLC-grade water, and mobile phase A was 0.1% formic acid in HPLC-grade water. The mass-to-charge ratio (*m/z*) MS scan range was *m/z* 250–2500, and MS/MS scan range was *m/z* 100–2500.

#### 2.3.2. Generation of Peptide Ion Library

Data for 12 fractioned peptide samples were acquired via IDA and imported into Protein Pilot 5.0.2. software, and processed using the Paragon protein database search algorithm (AB Sciex, Concord) to generate a peptide ion library. The Uniprot human database was used to match each MS/MS spectrum. The parameters for constructing the library were as follows: Alkylation with iodoacetamide, tryptic digestion, no special factors, >95% confidence, and select run false discovery rate (FDR) analysis. The generated library comprised of 141,916 spectra, 8090 peptides, and 239 proteins with 1% FDR.

#### 2.3.3. Data Independent Acquisition (DIA) and Relative Quantification

For relative quantification, samples were analyzed individually. In each run, 2 µL sample solution (0.5 µg/µL) was loaded onto an Ekigent ChromXP NanoLC trap column (350 µm i.d. × 0.5 mm, ChromXP C18 3 µm) at a flow rate of 5000 nL/min for 5 min. The injected sampled was eluted from the Eksigent Chrom XP NanoLC column (75 µm i.d. × 15 cm), at a flow rate of 300 nL/min for 120 min with a gradient of 5–90% mobile phase B. The following LC gradient was used: (time/mobile phases B%) 0 min/5%, 10.5 min/40%, 111.5 min/90%, 112 min/5%, and 120 min/5%. Mobile phase B was 90% acetonitrile/0.1% formic acid in HPLC-grade water, and mobile phase A was 0.1% formic acid in HPLC-grade water. The MS scan range was *m/z* 250–2500, and MS/MS scan range was *m/z* 100–2500. Peakview software (AB Sciex) was used to analyze DIA data for relative quantification. The parameters for data processing were as follows: Eight peptides per protein, five transitions per peptide, 99% threshold of peptide confidence, and 1.0% FDR threshold. For further statistical analysis, the total area sums method was used for normalization.

#### 2.3.4. Absolute Quantification Using Multiple Reaction Monitoring (MRM)

Skyline software (http://proteome.gs.washington.edu/software/skyline; accessed on 31 August 2020) was used to select the optimal charge state and fragmented ions for the acquisition method of each peptide. The gene name and peptide sequence were entered into Skyline software and the precursor of each peptide that was double-charged with three fragment ions was selected. Using a mixture of peptides, the collision energy was determined via a direct fusion experiment. To determine the collision energy, declustering potential and collision exit potential compound optimization of MRM analysis were performed with both Q1 and Q3 pairs by MRM scanning. For absolute quantification of each patient sample, a SCIEX Exion LC and QTRAP 5500 were used. The amount of the sample used in the analysis was 5 µL. An ACQUITY UPLC BEH C18 VanGuard pre-column (130 Å, 1.7 µm, 2.1 × 5 mm) and ACQUITY UPLC BEH C18 column (130 Å, 1.7 µm, 2.1 × 150 mm) at a flow rate of 250 µL/min and 30 min of total running time were used for analysis. The following settings were used over a 30-min gradient: 1 min, 5% B; 20 min, 40% B; 25 min, 90% B; and 30 min, 5% B. The mobile phases were the same as those used for DIA analysis (A: 0.1% FA in water, B: 0.1% FA in acetonitrile). The settings for the acquisition method were as follows: Source parameters, curtain gas (30 psi), low collision gas, ion spray (5500 V, 400 °C), ion source gas 1 (40 psi), and ion source 2 (60 psi). Peptides were synthesized using the standard chemicals and purity was more than 90% (Peptron, Daejeon, Korea). The standard curve including the concentration of each sample with more than five continuous data points were used.

### 2.4. Statistical Analysis

MarkerView (AB Sciex) and GraphPad Prism software (La Jolla, CA, USA) were used to perform statistical analysis. Principal component analysis and *t*-test were used to analyze peptides with significantly altered abundance. STRING 11.0 (https://string-db.org/; accessed on 1 January 2021) software was used to analyze protein-protein interaction networks and for functional enrichment.

## 3. Results

### 3.1. Protein Identification and Candidate Selection

Label-free quantification using SWATH was performed to analyze proteins in the serum collected from patients with MDD in depression and remission states. The SWATH results were matched to a peptide ion library, and 252 proteins were identified using MarkerView software (AB Sciex). The total area sums were used to normalize the data before statistical analysis. The *t*-test was performed to identify significantly differentially expressed proteins (*p* < 0.05). Twenty-two proteins were differentially expressed by more than 1.2-fold. From each sample, all peptides were checked using two criteria: FDR less than 1 and intensity more than 1000. Proteins that did not meet the criteria were not selected as candidates. Seven of the remaining proteins ultimately selected as candidates were alpha-1-acid glycoprotein 1, antithrombin-III, apolipoprotein C-II, complement component C8 gamma chain, lumican, prothrombin, and serum amyloid A-4 protein (Table 2). Heatmap analysis was performed using these seven candidates (Figure 2).

### 3.2. Pathway and Gene Ontology (GO) Analysis of Selected Differentially Expressed Proteins (DEPs)

Pathway and GO analyses were conducted using the seven candidate proteins using STRING 11.0 (Figure 3). The result of reactome pathway analysis showed that the formation of fibrin clot, regulation of complement cascade, regulation of IGF transport and uptake by IGFBPs, hemostasis, platelet activation, signaling, aggregation, and innate immune system are related to MDD (Figure 3a). GO process analysis further revealed that acute inflammatory response, acute-phase response, regulation of protein activation cascade, inflammatory response, and leukocyte-mediated immunity are associated with MDD (Figure 3b).

### 3.3. Peptide Selection for MRM Analysis

MRM analysis was conducted for the seven candidates. To analyze targeting peptides, one peptide from each candidate was selected: SDVVYTDWK (alpha-1-acid glycoprotein 1), TSDQIHFFFAK (antithrombin-III), ESLSSYWESAK (apolipoprotein C-II), SLPVSDSVLSGFEQR (complement component C8 gamma chain), SLEYDLSFNQIAR (lumican), ELLESYIDGR (prothrombin), and FRPDGLPK (serum amyloid A-4 protein). Five criteria were used to select the peptide: (1) Peptide length of 7–15 amino acids; (2) sequence without mis-cleavage site; (3) sequence without modifications; (4) sequence without M; and (5) FDR of peptide <1. Skyline software (http://proteome.gs.washington.edu/software/skyline; accessed on 31 August 2020) was used to present the transition information of selected peptides. The results of the calibration curves of all peptides are shown in Appendix A.

### 3.4. Candidate Validation by MRM

The selected protein candidates were confirmed by absolute quantification in MRM. Ultra-performance liquid chromatography (UPLC)-MRM/MS was used for MRM analysis. Five microliters of the sample were injected and analyzed. For optimization, three Q3 ions were selected for MRM analysis. Each sample was quantified using only the Q3 ion that showed the highest sensitivity (Table 3). Statistical analysis was performed using MRM data. Normality testing, t-test, and receiver operating characteristic (ROC) analysis were performed. The final candidates selected were those with the same tendency of protein expression in both SWATH and MRM data, and those with a *p*-value < 0.05. Prothrombin was the only protein that met both criteria. Thus, prothrombin was selected as the final biomarker candidate for MDD (Figure 4a). The area under the curve (AUC) was 0.66 (Figure 4b).

## 4. Discussion

To identify the biomarkers of depression, proteomic analysis was performed on the serum samples obtained from the patients with depression and in remission state; proteins significantly and differentially expressed in these samples, with fold-change difference of more than 1.2, were first selected. Subsequently, only proteins with an intensity of ≥1000 and FDR ≤ 1 of the candidate protein were selected. Next, significant proteins were selected as the final protein markers during the validation process using MRM. The only protein showing a significant difference between the two statuses was prothrombin.

Prothrombin is a precursor of thrombin. In the blood coagulation pathway, thrombin converts factor XI to XIa, VIII to VIIIa, V to Va, fibrinogen to fibrin, and XIII to XIIIa. It also promotes platelet activation and aggregation through protease-activated receptor (PAR) activation on the platelet surface. PARs form a thrombin receptor family [27,28], which is a unique family of G-protein coupled receptors. PAR expression is high in platelets, neurons, endothelial cells, and muscle cells. Thrombin can signal through PAR1, PAR3, and PAR4 [29]. Thrombin can enter the brain by crossing the blood-brain barrier via hyperpermeability [30] and can also be formed from prothrombin locally in the brain [31]. PAR1 is expressed on the astrocytes of white and gray matter [32]. Activated PAR1 in microglia induces an increase in the levels of intracellular Ca^2+^ and mitogen-activated protein kinases (MAPKs) p38 and p44/42 [33]. The expression of four genes involved in the MAPK pathway (*Mkp-1, p38, Pkc,* and *Erk*) was significantly increased in depressed mice in a previous study [34]. Prothrombin expression can be controlled by p38-MAPK pathway, which regulates processing of the 3′ end of RNA [35]. Additionally, PAR1 activation stimulates thrombin-induced microglial proliferation [36], which can subsequently impair hippocampal neurogenesis, and the spatial memory ability in mice [37]. Moreover, PAR1 can transactivate PAR2, which leads to long-term depression [38].

Previous studies have shown that platelet activation is increased in MDD patients [39,40,41]. Activation of PAR1 and PAR4 triggers platelet secretion and aggregation [42]. Brain-derived neurotrophic factor and serotonin, which are related to MDD, are secreted by platelets [43]. Meanwhile, treatment with selective serotonin reuptake inhibitors, such as sertraline and paroxetine, can decrease platelet activation [44,45]. This indicates that increased platelet activation is related to depressive symptoms. These results support the hypothesis that serotonin depletion is related to MDD. Platelet aggregation further leads to thrombosis, which causes coronary artery diseases and MDD is a well-known risk factor for this diseases [46,47,48]. Hence, selective serotonin reuptake inhibitors may protect against cardiovascular events [49].

In this study, we compared the proteomic profiles of patients with depression and remission statuses. Prothrombin is a biomarker that can distinguish the depression status from the remission status. Considering that both groups of patients were administered antidepressants, prothrombin may also be used for monitoring the disease status and function as an effective prognosis biomarker. MDD remission occurs when a patient has recovered from the disease and has not relapsed. Therefore, the remission status of patients with MDD likely represents healthy controls. However, it is necessary to confirm the accuracy of prothrombin as a biomarker by evaluating a healthy control group. Additionally, including a larger sample size, comparative analysis of different subgroups of patients with MDD should be completed, along with the healthy controls. Moreover, it would be interesting to examine samples from the same patient during different MDD statuses to eliminate variables and confirm the differences in biomarker expression within individuals. Finally, pathway analysis of thrombin-induced platelet activation and MDD is required to gain an etiological understanding of MDD. Thus, prothrombin is a candidate powerful biomarker for diagnosing and predicting the prognosis of MDD.

## 5. Conclusions

Prothrombin was identified as a novel biomarker using sera from MDD patients in depression and remission statuses. Candidates were identified using the SWATH acquisition method and validated by MRM analysis. Prothrombin was the only protein selected as a biomarker for MDD. Moreover, increased prothrombin levels were related to platelet activation via activation of PARs. However, these results must be validated in a future study with a larger sample size, as the AUC of ROC was 0.6. Nevertheless, our study provides reliable preliminary results as the actual concentration of each peptide was measured by relative and absolute quantification using mass spectrometry. Thus, prothrombin represents an effective screening and prognostic candidate biomarker for MDD.

## Figures and Tables

**Figure 1 diagnostics-11-00539-f001:**
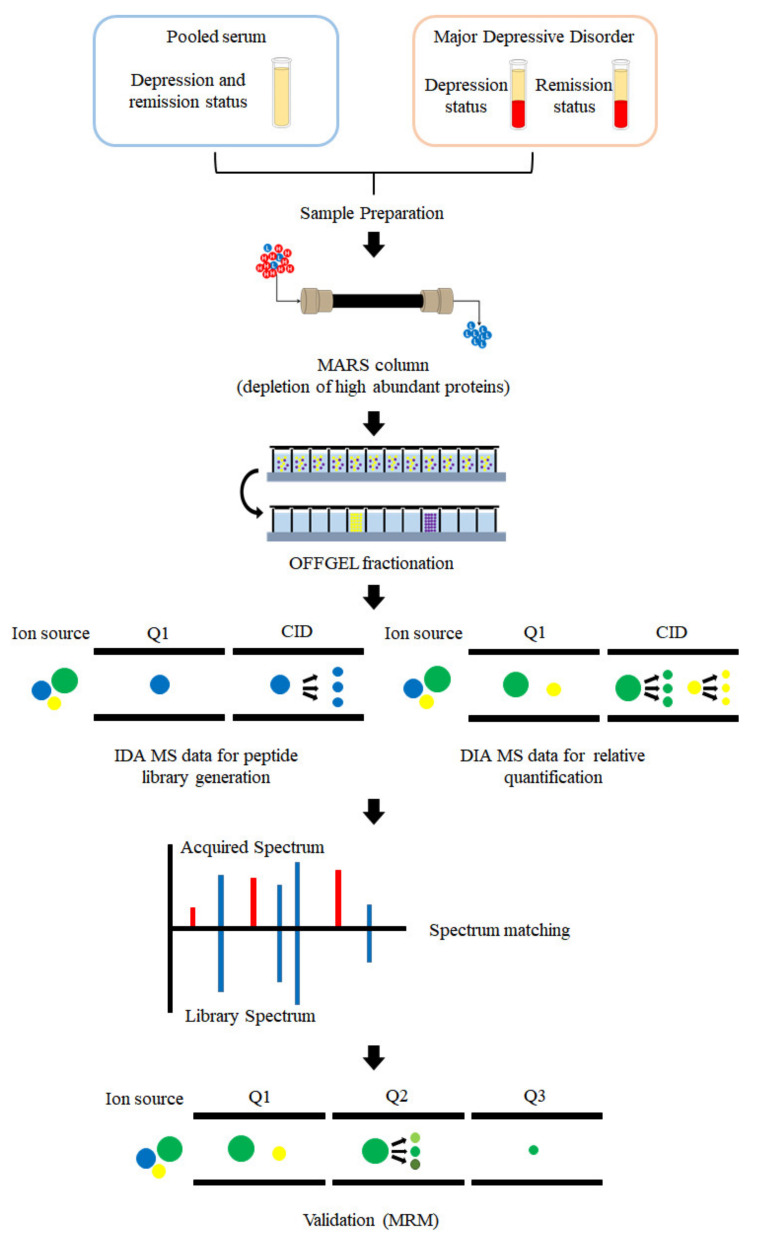
Experimental scheme for discovering the biomarkers for major depressive disorder (MDD). Blood was collected from patients with MDD and those in remission. Pooled serum was collected from all patients with MDD and those in remission. High abundant proteins in all samples including pooled serum were depleted using a multiple affinity removal system (MARS) LC column. Low-abundant proteins were digested with trypsin. The pooled serum was fractionated by OFFGEL electrophoresis and used to generate a peptide library by information-dependent acquisition MS analysis; each sample was analyzed in data independent acquisition (DIA)-MS mode. The DIA-MS data acquired from each sample were matched with the peptide library. For validation, multiple reaction monitoring (MRM) mode was used for absolute quantification of the selected peptide.

**Figure 2 diagnostics-11-00539-f002:**
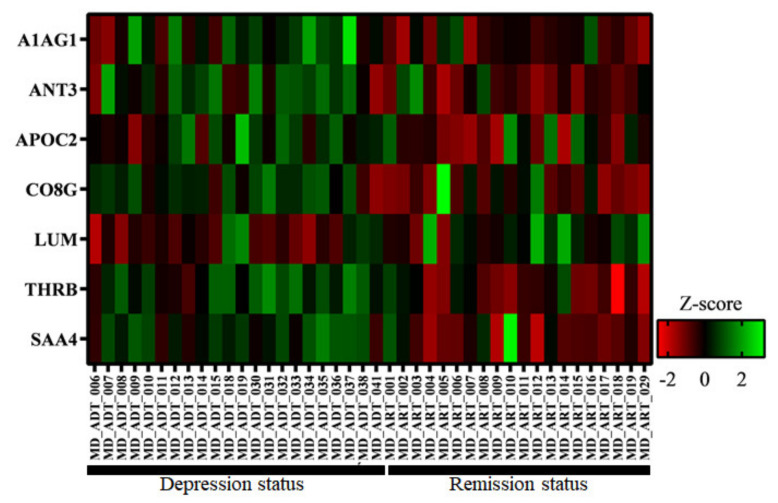
Differentially expressed proteins from patients with depression status compared to those with remission status. Proteins were significantly differentially expressed by more than 1.2-fold as shown using z-score. *X*-axis shows the samples used for analysis. ADT refers to depression group and ART refers to remission group. *Y*-axis shows the seven biomarker candidates from IDA-SWATH analysis. A1AG1: Alpha-1-acid glycoprotein 1, ANT3: Antithrombin-III, APOC2: ApolipoproteinC-II, CO8G: Complement component C8 gamma chain, LUM: Lumican, THRB: Prothrombin, SAA4: Serum amyloid A-4 protein.

**Figure 3 diagnostics-11-00539-f003:**
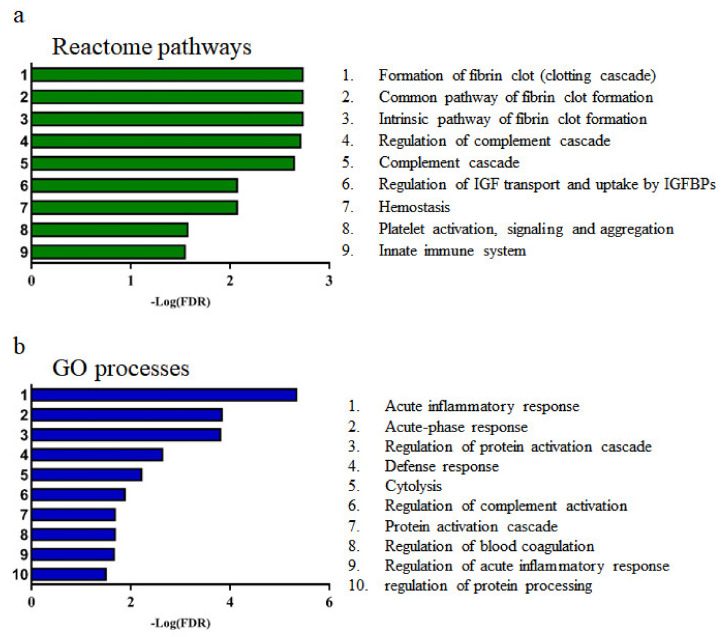
Reactome pathways and GO processes associated with proteins that are differently expressed in depression status and remission status. (**a**) Results of reactome pathways related to candidate proteins. (**b**) Results of GO processes related to candidate proteins. The X and Y axes represent the –log (FDR) and ranking of the lower FDR, respectively. The lower FDR denotes a strong association with MDD and its remission.

**Figure 4 diagnostics-11-00539-f004:**
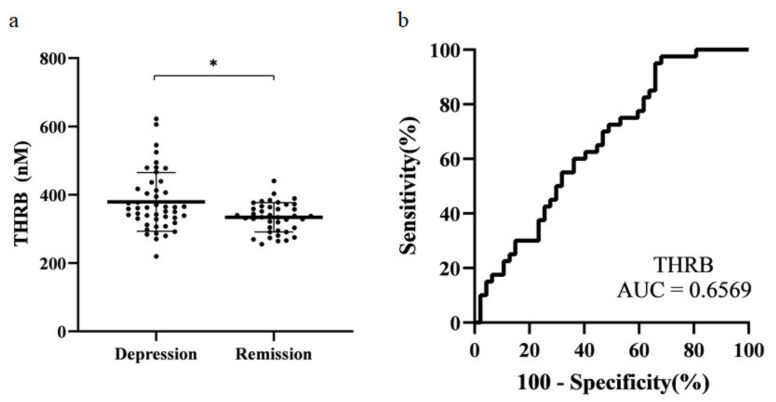
Scatter plot and receiver operating characteristic (ROC) analysis curve for prothrombin. (**a**) Prothrombin concentration in each subject. The plot shows mean ± standard deviation. * *p* < 0.05. (**b**) ROC of prothrombin (AUC = 0.66). THRB: Prothrombin.

**Table 1 diagnostics-11-00539-t001:** Study subject characteristics.

Variable	Discovery Set		Validation Set	
	Depression Status (*n* = 22)	Remission Status (*n* = 20)	Depression Status (*n* = 47)	Remission Status (*n* = 40)
Sex (female/male)	19/3	17/3	40/7	34/6
Age (years)	47.64 ± 14.81	50.60 ± 12.46	50.04 ± 17.85	61.18 ± 15.00
BMI	22.6 ± 2.19	24.51 ± 3.72	23.53 ± 3.23	24.29 ± 3.33
BDI	30.18 ± 12.53	7.10 ± 6.95	27.30 ± 12.48	8.00 ± 7.28
HAMD	21.18 ± 6.84	3.25 ± 3.43	16.51 ± 8.15	4.5 ± 4.64
Smoking (yes/no)	3/19	3/17	5/40	4/34

BMI: Body mass index, BDI: Beck depression inventory, HAMD: Hamilton depression rating scale.

**Table 2 diagnostics-11-00539-t002:** Candidate proteins with associated *p*-value and fold-change.

No.	Compound Name	Uniprot ID	Gene name	*p*-Value	Fold Change *
1	Alpha-1-acid glycoprotein 1	P02763	*ORM1*	5 × 10^−3^	1.29
2	Antithrombin-III	P01008	*SERPINC1*	2 × 10^−3^	1.30
3	Apolipoprotein C-II	P02655	*APOC2*	0.05	1.24
4	Complement component C8 gamma chain	P07360	*CO8G*	0.01	1.36
5	Lumican	P51884	*LUM*	0.01	0.82
6	Prothrombin	P00734	*F2*	1 × 10^−5^	1.25
7	Serum amyloid A-4 protein	P35542	*SAA4*	1 × 10^−3^	1.22

* Depression status vs. remission status.

**Table 3 diagnostics-11-00539-t003:** Peptide parameters for MRM.

Compound Name	Gene Name	UniProt ID	Peptide Sequence	Ion Charge State	Q1(*m/z*)	Q3(*m/z*)	Q3 Ion Type	DP (Volts)	CE (Volts)	CXP (Volts)
Alpha-1-acid glycoprotein 1	*ORM1*	P02763	SDVVYTDWK	**2**	**556.767**	**811.398**	**y6**	**71.7**	**28.9**	11.0
					556.767	712.330	y5	71.7	28.9	11.0
					556.767	302.135	b3	71.7	28.9	11.0
Antithrombin-III	*SERPINC1*	P01008	TSDQIHFFFAK	**3**	**447.559**	**189.087**	**b2**	**63.7**	**21.9**	**11.0**
					447.559	796.414	y6	63.7	21.9	11.0
					447.559	659.355	y5	63.7	21.9	11.0
Apolipoprotein C-II	*APOC2*	P02655	ESLSSYWESAK	**2**	**643.799**	**870.399**	**y7**	**78.0**	**32.0**	**11.0**
					643.799	957.431	y8	78.0	32.0	11.0
					643.799	620.304	y5	78.0	32.0	11.0
Complement component C8 gamma chain	*C8G*	P07360	SLPVSDSVLSGFEQR	**2**	**810.915**	**836.396**	**y7**	**90.2**	**35.0**	**11.0**
				3	540.946	723.312	y6	70.6	24.0	11.0
					540.946	201.073	b2	70.6	22.0	11.0
Lumican	*LUM*	P51884	SLEYDLSFNQIAR	**2**	**778.391**	**948.526**	**y8**	**87.9**	**36.9**	**11.0**
					778.391	1063.553	y9	87.9	36.9	11.0
					778.391	835.442	y7	87.9	36.9	11.0
Prothrombin	*F2*	P00734	ELLESYIDGR	**2**	**597.804**	**839.389**	**y7**	**74.7**	**30.4**	**11.0**
					597.804	710.347	y6	74.7	30.4	11.0
					597.804	243.134	b2	74.7	30.4	11.0
Serum amyloid A-4 protein	*SAA4*	P35542	FRPDGLPK	**2**	**465.264**	**686.362**	**b6**	**65.0**	**25.6**	**11.0**
					465.264	244.166	y2	65.0	25.6	11.0
					465.264	573.278	b5	65.0	25.6	11.0

The bolded ion type was used in final quantification. DP: Declustering potential; CE: Collision energy; CXP: Collision exit potential.

## Data Availability

The data presented in this study are available on request from the corresponding author. The data are not publicly available due to personalized data of patients included to this study and due to the medical confidentiality.

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
