# Peer review of "Discovery of Screening Biomarkers for Major Depressive Disorder in Remission by Proteomic Approach"

_diagnostics, 2021, doi:10.3390/diagnostics11030539_

Round 1

Reviewer 1 Report

This manuscript reported the discovery of biomarkers for major depressive disorder in remission by proteomic approach, the presentation of the results is very clear, and the content is well organized, however, some technical issues should be explained.

  1. This manuscript used IDA mode to build the library, although they depleted high abundance proteins, only 239 proteins were identified (DDA mode 500+), one reason is the flow rate which is not nano-LC, the system will be more stable when higher flow rate is used. The explanation should be added.
  2. Table 3, the charge of the peptide should be added since multiple charged peaks are observed for the peptide. Q3 ion type, 2y6, this expression is not standard, what does 2 mean? How about the charge? footnote should be added.

    3. MRM is used for the absolute quantification of protein, some results             are missing, such as the calibration curve of each peptide, this can be           put in SI.  

Reviewer 2 Report

This paper provides a high-quality proteomic screen for biomarkers for Major Depressive Disorder, by combining a quantitative DIA proteomics approach (SWATH method) with an absolute quantification validation using the Multiple Reaction Monitoring method. The authors approach a question of great medical interest with an appropriate experimental design. The manuscript is well written and provides a potential biomarker (prothrombin) that will require further validation to confirm these results.

A few comments/questions,
              1 – I would appreciate in Figure 4a) a description of what is exactly plotted in this Figure. In addition to the data, is it the mean that is indicated? And the error bar corresponds to standard deviation, standard error or a specific quartile?
              2 – I would ask the authors for a more detailed description of how prothrombin could be used to monitor the disease status or as a prognosis biomarker. Although there is a significant difference in mean as indicated by the t-test, the distribution of prothrombin concentrations are highly overlapping for both conditions; a big portion of the patients with depression display prothrombin concentrations that are already in the range observed for patients in remission.

Round 2

Reviewer 1 Report

In supplementary information, calibration curve for b is missing. 

Author Response

Thank you for your carefully reviewing our confirmation of our manuscript and supplementary material. We checked the uploaded file and we didn't find nothing missing in supplementary material. But to be sure, we will upload  supplementary material again. Thank you.